# Developing an Olive Biorefinery in Slovenia: Analysis of Phenolic Compounds Found in Olive Mill Pomace and Wastewater

**DOI:** 10.3390/molecules26010007

**Published:** 2020-12-22

**Authors:** Ana Miklavčič Višnjevec, Paul Baker, Adam Charlton, Dave Preskett, Kelly Peeters, Črtomir Tavzes, Katja Kramberger, Matthew Schwarzkopf

**Affiliations:** 1Faculty of Mathematics, Natural Sciences and Information Technologies, University of Primorska, Glagoljaška 8, SI-6000 Koper, Slovenia; ana.miklavcic@famnit.upr.si (A.M.V.); crtomir.tavzes@famnit.upr.si (Č.T.); 2The Biocomposites Centre, Bangor University, Bangor, Gwynedd LL57 2DG, UK; paul.baker@bangor.ac.uk (P.B.); adam.charlton@bangor.ac.uk (A.C.); d.preskett@bangor.ac.uk (D.P.); 3InnoRenew CoE, Livade 6, SI-6310 Izola, Slovenia; kelly.peeters@innorenew.eu; 4Faculty of Health Sciences, University of Primorska, Polje 42, SI-6310 Izola, Slovenia; katja.kramberger@fvz.upr.si

**Keywords:** *Olea oleuropea* L., olive mill effluents, pomace, HPLC-DAD-qTOF, phenolic compounds

## Abstract

The valorization of olive pomace through the extraction of phenolic compounds at an industrial scale is influenced by several factors that can have a significant impact on the feasibility of this approach. These include the types and levels of phenolic compounds that are present, the impact that seasonal variation and cultivar type have on the phenolic compound content in both olive pomace and mill effluents and the technological approach used to process the olive crop. Chemical analysis of phenolic compounds was performed using an HPLC-diode-array detector (DAD)-qTOF system, resulting in the identification of 45 compounds in olive mill wastewater and pomace, where secoiridoids comprised 50–60% of the total phenolic content. This study examined three different factors that could impact the phenolic compound content of these processing streams, including cultivar types typically grown on local farms in Slovenia, the type of downstream processing used and seasonality effects. Olive crop varieties sourced from local farms showed high variability, and the highest phenolic content was associated with the local variety “Istrska Belica”. During processing, the phenolic content was on average approximately 50% higher during two-phase decanting compared to three-phase decanting and the type of compound present significantly different. An investigation into the seasonal effects revealed that the phenolic content was 20% higher during the 2019 growing season compared to 2018. A larger sample size over additional growing seasons is required to fully understand the annual variation in phenolic compound content. The methods and results used in this study provide a basis for further analysis of phenolic compounds present in the European Union’s olive crop processing residues and will inform techno-economic modelling for the development of olive biorefineries in Slovenia.

## 1. Introduction

The production of olive oil in the Istrian region of Slovenia has a long-established tradition dating back to the 4th Century BC [1]. At the heart of this is the “Istrska belica” cultivar of olives (Istrian white olives), which have been praised for their ability to withstand low temperatures during cultivation, maximizing oil content and possessing an excellent taste. This cultivar has high levels of monounsaturated fatty acids and biologically active molecules, including phenolic compounds, squalene and tocopherols [2,3,4,5], in comparison to other Slovenian Istria varities, which contribute to the organoleptic profile of the oil produced from these olives [6]. Phenolic compounds from olives may offer a variety of benefits to human health, including a reduction in coronary heart disease risk factors, prevention of several types of cancers and modification of immune and inflammatory responses [7,8,9]. However, the full clinical potential of these compounds is still to be fully quantified and may be affected by bio-accessibility, based on absorption and colonic fermentation, distribution and metabolism [5,10].

Modern, industrial olive oil extraction uses a continuous process in which the oil is separated from the olives using two- or three-phase decanting centrifugation. The two-phase decanter centrifuge generates a waste called alperujo, which is a mixture of pomace, oil and water, whilst the three-phase decanter produces relatively low moisture pomace and olive mill wastewater (OMWW). The pomace contains the remaining olive pulp, skin, stones and water [11,12,13]. A destoning process can be incorporated into the process leading to the removal of 70% of the stones. While there are many valuable compounds still present in the pomace [10,14,15,16], successful and economically viable extraction methods are still in development. Rubio-Senent et al. [16] investigated the possibility of obtaining simple phenolic compounds in high yield from two-phase olive waste using a series of hydrothermal treatments and concluded that phenolic extracts in the aqueous and lipid system inhibited oxidation better compared to the untreated control.

Olive pomace is currently used in a number of applications as fertilizer, compost, animal feed or for bioenergy [14], but some integrated biorefinery approaches for higher value applications have also been proposed [17,18]. Romaro-Garcia [17] proposed ethanol production as one of most promising applications as well as obtaining antioxidants, oligosaccharides and lignin. Schievano et al. [18] proposed a biorefinery approach for pomace valorization, using supercritical carbon dioxide, coupled with polar solvents, to produce value-added products such as antioxidants, biofuels, energy or sustainable sources of carbon for soil. The authors concluded that further work, including the energy balance and evaluation of the market values of the extracts, was required in order to assess the real feasibility of the proposed approach.

Olive mill wastewater (OMWW) is the processing water obtained from the three-phase decanting method, and is acidic with high levels of organic pollutants [17]. There are currently few uses for this effluent due to variability in the composition and current process limitations include the handling of large volumes of material and degradation of the effluent during downstream processing as a result of oxidation. The high concentration of phenolic compounds from OMWW, produced during processing, can also have a severe environmental impact if they are improperly released into local waterways. It is therefore important from an economic perspective, to establish the feasibility of recovering phenolic compounds at industrial scale from the olive crop processing streams, including pomace and OMWW and to determine impact of key factors, including seasonality, on the potential yields of these important molecules.

More than 50 different phenolic compounds have been identified in both olive pomace and stones and OMWW also contains simple phenolic compounds, benzoic and cinnamic acids derivatives, flavonoids, lignans and secoiridoids [19], with the latter molecules found specifically in olives [20,21]. During the olive oil manufacturing process, ligstroside and oleuropein can enter different transformation-reaction pathways involving plant enzymatic and chemical transformation [22]. When the transformation pathway reaches its end and the olive oil has already lost its freshness and antioxidative properties after one or two years of storage, the total phenolic compounds content can be relatively high with higher amounts of simple phenolic compounds such as tyrosol and hydroxytyrosol [6]. The same process of complex phenolic compounds breaking down into simple phenolic compounds, such as tyrosol and hydroxytyrosol, is expected to occur in olive mill effluents over relatively short periods of time. Therefore, it is important to identify each phenolic compound, rather than total phenolic content, in order to evaluate the level of phenolic breakdown.

The aim of this study was to identify and quantify the phenolic compounds in OMWW and pomace generated from the commercial processing of material to extract olive oil. The first level of variation occurs at the local farms in Slovenian Istria where different varieties of olive crops, such as “Istrska belica”, “Leccino”, “Buga” and “Maurino” are grown. The second level of variation occurs during processing when different decanting technologies are used to recover the oil. Finally, the third level of variation occurs across different growing seasons. This is the first comprehensive report that has evaluated all three of these important parameters in an industrial context, in order to assess the potential for the valorization of olive residues.

## 2. Results and Discussion

### 2.1. Identification of Phenolic Compounds in Olive Mill Wastewater and Pomace

The following groups of phenolic compounds were identified in pomace and in OMWW: (1) simple phenolic compounds; (2) benzoic acids; (3) cinnamic acid and (4) secoiridoids.

The presence of tyrosol was confirmed in olive pomace and olive mill water by reference to the retention time of a standard solution by only diode-array detector (DAD) detection due to the fact that tyrosol cannot be detected by MS because of its high ionization energy [19]. The compounds that were confirmed through reference to a standard solution by exact mass, fragmentation pattern and retention time were hydroxytyrosol, luteolin, verbascoside, vanillin, apigenin and oleuropein. All the other compounds were identified based on the exact mass and fragmentation pattern already determined in the literature [19,23,24,25,26,27,28,29,30,31,32,33,34,35,36,37,38]. In addition, the UV absorption maxima of the noted peak was determined.

Similar phenolic compounds were identified both in pomace and olive mill water (Table 1) with the exception of oleoside, elenolic acid glucoside, verbascoside, nuzhenide, oleuropein, oleuropein-aglycone dialdehydes (3,4-DHPEA-EDA), *p*-HPEA-EDA, and coumaric acid glucoside that were only identified in pomace but not in olive mill water (Figure 1).

Previous reports [19,23,30] have described the presence of four peaks with the exact mass of oleoside, and a fragmentation pattern characteristic for oleoside was found at relative retention times 0.92 (peak number 1), 0.96 (peak number 2), 1.00 (peak number 3) and 1.21 (peak number 7) in olive mill pomace. The four peaks had slightly different fragmentation profiles (Appendix A). For the first two peaks determined at 0.92 (peak number 1) and 0.96 (peak number 2) the non-typical UV absorption maxima was observed as previously reported [19]. However, the third and fourth peaks include typical absorption maxima at 230 nm. In this study it was possible to confirm the previously observed co-elution by Jerman Klen et al. [19] of the oleoside (peak number 3) at 1.00 relative retention time with hydroxytyrosol, and the tentative identification of secologanoside, due to absorption maximum at 230 nm and the highest abundance of the fragments 389 and 345. A tentative identification of secologanoside in olive pomace and OMWW was made, in accordance with a previous report [19].

Elenolic acid glucoside was previously reported in olive oil process derived matrices, including leaves [23,30,33], olive fruits [19,34,35], olive oil [19], pomace [19,36,37] and OMWW [20]. Four different isomers of elenolic acid glucoside have been tentatively identified previously in pomace, but not all four were identified in OMWW [19,23]. In all isomers, the fragments 403, 223 and 179 were found as previously reported [19,23]. The fragment with *m*/*z* to 223 corresponds to the elimination of hexose, giving rise to *m*/*z* 179 by the neutral loss of CO_2_ [19].

A previous report [19] detected verbascoside in OMWW, but our studies were not able to confirm the presence of this compound in any of the OMWW samples that were analyzed. As previously reported during studies on olive fruits [25], verbascoside may exist as a pair of geometric isomers arising from the caffeic acid moiety or different attachment of the sugar to the aglycone. The presence of verbascoside was confirmed through comparison with the retention time of a standard solution (peak number 14, Figure 1), similar to two β-OH-verbascoside isomers that were found in both pomace and olive mill water (Appendix A). At relative retation time of 1.56 (peak number 17), a possible verbascoside isomer was identified that was present in a higher amount compared to the peak number 14 (Table 1).

On the basis of mass accuracy and fragmentation patterns (Isomer 1: 523, 685, 453, 421, 299 and 223; Isomer 2: 523, 685, 453, 299 and 223), two different isomers of nuzhenide were identified in olive pomace but not in OMWW, which is in agreement with previous reports [35,38]. Previously, these compounds have only been found in olive stones [38]; therefore, it is likely that some of the stones were crushed during processing and ended up in the pomace fraction.

The presence of oleuropein was identified by a pure standard at relative retention time 1.79 (peak number 24). At relative retention times 1.87 (peak number 27) and 1.92 (peak number 28), two similar compounds were tentatively identified as oleuropein isomers with *m*/*z* 539 and similar fragmentation patterns as the oleuropein pure standard [23]. The last eluted oleuropein isomer was present in OMWW as well.

Oleuropein-aglycone dialdehydes (3,4-DHPEA-EDA) with exact molar masses of 319.1185 (Isomer 1) and 319.1187 (Isomer 2) were tentatively identified at relative retention times 1.83 (peak number 25) and 2.17 (peak number 35) with similar fragmentation patterns as previously reported [19].

*p*-HPEA-EDA (or oleocanthal) has one hydroxyl group less than 3,4-DHPEA-EDA as reported by Cioffi et al. [31]. A similar retention time and fragmentation pattern for 3,4-DHPEA-EDA was determined from our studies and as previously reported [19,32].

Trans *p*-coumaric acid 4-glucoside was identified in pomace by exact mass detecting fragments 163 and 119, and in accordance with a previous report [19]. The same fragmentation pattern for *p*-coumaric acid was previously reported by Araújo et al. [28].

### 2.2. Quantification of Phenolic Compounds in Pomace

The mean ± standard deviation, median, minimum and maximum levels of individual, total phenolic compounds and different groups of phenolic compounds, such as simple phenolic compounds, benzoic acids, cinnamic acid, flavonoids and secoiridoids, together with the Spearman Rank correlation coefficients between phenolic compounds levels and scavenging activity measured using DPPH assay in all pomace samples are shown in Table 1. All results are expressed as mg/kg dry weight (dry wt) of pomace sample. Although from the literature it is well known that the phenolic compound concentrations are affected by agronomic and technological factors, including the cultivar type, ripening stage and geographic origin [6,31], the total amount of phenolic compounds that varied from 851 mg/kg dry wt to 4473 mg/kg dry wt (Table 1) are in the range previously reported elsewhere [14,31]. Podgornik et al. [14], with the same extraction and detection method, determined that the total phenolic compounds range from 500 to 8000 mg/kg d.w. The wide variation of phenolic compounds is consistent with the literature, with the highest levels of total phenolic compounds found in samples from the variety “Istrska belica” (two-phase decanter). The main group of phenolic compounds in pomace was secoiridoids that comprised on average 71 ± 7%, with the 3,4-DHPEA-EDA and oleuropein or oleuroside that are elute at the same time being the most abundant. A previous report determined that 50–70% of the total phenolic content in olive pomace from the same region was attributed to secoiridoids [31]. In contrast to our results, Madureira et al. [39] determined that the major compound present was hydroxytyrosol in extracted and crude pomace from Portugal. This might be due to both differences in the cultivar types screened and environmental factors in the two studies, which would seem to indicate the importance of the evaluation of the phenolic compound content in relation to geographic location and the olive varieties cultivated. In addition, the differences might be due to different post-harvested treatments and storage of the samples. These phenolic compounds could have a range of useful applications from use as therapeutic agents to control colorectal cancer [40], to ingredients for the production of fortified food [41,42]. In contrast to a previous report [43], where simple phenolic compounds were determined as the main compounds in pomace, both tyrosol and hydroxytyrosol were present at 8 ± 5% of total phenolic compounds in the samples analyzed for this study. The low amounts of simple phenolic compounds and the majority of complex phenolic compounds, such as secoiridoids, identified in our study are promising for potential industrial end-users (e.g., cosmetics and personal care) in applications where antioxidant activity of the extracts is very important [17]. These simple phenolic compounds may be the primary products from the oxidative decomposition of secoiridoids [44,45]. In our previous study [6], it was determined that an increase in tyrosol and hydroxytyrosol and decrease in secoiridoids levels resulted after one and two years of storage for extra virgin olive oil samples.

There are several possible factors causing variation levels in the level of phenolic compounds in OMWW and pomace generated from olive oil extraction industrial processes, as highlighted. However, from the current state-of-the-art industrial point of view (often very difficult to control the input crop) and from a preliminary statistical analysis, the variation according to different growing seasons and different separation (centrifugation) technologies that are discussed in Section 2.2.1 and Section 2.2.2 were chosen.

#### 2.2.1. Variation in Phenolic Compound Content in Olive Pomace across Different Growing Seasons

Phenolic compounds are secondary plant metabolites and are synthesized in response to environmental stress factors, including microbial attack, tissue damage, UV rays [46] and water deficiency in olives, resulting in increased concentrations of these molecules [47]. In general, extreme weather conditions can significantly influence the concentrations of phenolic compounds, and it has been determined that the increase in the level of these compounds in extra virgin olive oil, across three years (2011–2013), was strongly influenced by these factors. The oils contain the highest quantity of phenolic compounds in most crop years and in areas with the highest water deficiency [6].

In order to investigate and quantify any seasonal variation in phenolic compound levels for specific regions in Slovenia, pomace samples produced following separation using three-phase decantation were collected and investigated across two growing seasons (2018 and 2019). The differences in the levels of total phenolic compounds and the main groups of phenolic compounds determined in the pomace samples between the two years are shown in the Figure 2. These two crop years were chosen due to the variation in weather conditions. In contrast to 2018, the crop year 2019 was unusual; the yields were 50–60% lower in the region than previous years. The season began ten days earlier, and in the beginning of the season, the olives from the variety “Istrska belica” were also present, which is unusual because this is a late season variety. The unusual season was due to increased rainfall in the study region during certain periods of the year (May, July and September) [48], which allowed the development and spread of the olive fly that greatly affected the final olive yields.

It was determined that there were no statistically significant differences in total phenolic compounds, simple phenolic compounds, benzoic acids, cinnamic acids and secoiridoids content between the two years. The exception was the marginally significant differences (*p* = 0.05) in levels of flavonoids between the two years. In the case of crop year 2019 (median: 151 mg/kg dry wt), the levels of flavonoids in pomace samples were higher than in crop year 2018 (median: 108 mg/kg dry wt). In contrast to our study, Ntougias et al. [49] and Obied et al. [50] observed significant variation in phenolic compounds content between seasons in olive mill waste material. The fact that there were no observed significant differences between the two years (Figure 2) might be the consequence of different varieties, quality and maturity of olives present in the olive mill when the samples were taken. Analysis of a larger sample range would be necessary to observe the differences between the two years. However, the preliminary results in relation to the annual variation in phenolic compound levels in pomace samples are promising for further development of biorefinery in Slovenia due to the low variation observed between two crop years with very different weather conditions. In order to provide constant quality of raw material, it is necessary to be able to control the factors that influence variability. As more information about the additional growing season becomes available, the data will be updated.

#### 2.2.2. Variation in Phenolic Compound Content in Olive Pomace Using Different Separation (Centrifugation) Technologies

In contrast to the comparison in total phenolic compound content between crop years, statistically significant differences were observed when two different olive mill separation (centrifugation) technologies were compared (*p* = 0.037) (Figure 3).

The levels were higher in pomace samples taken from the two-phase decanting process (median: 2970 mg/kg dry wt), compared to the three-phase system (median: 1900 mg/kg dry wt), due to the addition of extra water to the olive paste in the latter process, which has a dilution effect and results in dissolved losses of phenolic compounds [51]. The two-phase decanting centrifuge process is an extraction system that is also known as “ecologic” or “water saving” because it does not require the addition of supplementary water, which reduces wastewater generation by up to 80%. The concept of working is similar to that of a three-phase decanter, except that a horizontal centrifuge has no, or reduced, requirement for additional water [11,12,13].

There were also significant differences between the main group of phenolic compounds present in pomace, secoiridoids (*p* = 0.0374), with a higher amount of pomace from the two-phase decanter (median: 1990 mg/kg dry wt) compared to three-phase separating decanter (median: 1270 mg/kg dry wt). In addition, significant differences were observed in vanillin content (*p* < 0.05) in pomace obtained following separation using the two-phase system (median: 43 mg/kg dry wt) compared to three-phase separating decanter (median: 6 mg/kg dry wt). The levels of other groups of phenolic compounds, including simple phenolic compounds, cinnamic acids and flavonoids, were not significantly different when the two separation technologies were compared.

This study indicates, for the first time, that the technological approach used in olive mills to separate the different fractions is a critical factor in determining the types and levels of phenolic compounds obtained in the resultant pomace.

#### 2.2.3. Radical Scavenging Activity by DPPH

The determination of radical scavenging activity, using the DPPH assay, is a suitable method for predicting the inhibition of primary oxidation product formation by natural plant extracts [47,48]. The EC50 value determined in the pomace samples correlates inversely with the concentrations of total phenolic compounds (r_s_ = −0.8; *p* < 0.05). The inverse correlation is expected because the EC50 value is defined as the concentration of substrate that causes 50% loss of DPPH activity (color) [52]. These calculations were made in order to evaluate the antioxidant activity of each of the phenolic compounds or group of the phenolic compounds in relation to their determined concentrations. However, it must be pointed out that the radical scavenging activity of phenolic compounds does not necessarily reflect the antioxidant activity of phenolic compounds in the cells in the organism.

The calculated Spearman Rank correlation coefficients reflect differently strong correlations between the levels of each phenolic compounds or group of the phenolic compounds and their radical scavenging activity by DPPH (Table 1). In some cases, the Spearman Rank Correlation coefficient was not significant. Therefore, no correlation was observed. As is shown in Table 1, the Spearman Rank correlation is the strongest between the total phenolic compounds and radical scavenging activity by DPPH (r_s_ = −0.81) as compared to the Spearman Rank correlation between each phenolic compound or groups of phenolic compounds determined in the samples and radical scavenging activity by DPPH (r_s_ varied from −0.60 to −0.77, where correlations were significant). This confirms the previously reported observation that the antioxidant pattern is usually complex, and it can include synergistic effects of the compounds that are not possible to determine by only the quantification of phenolic compounds by the HPLC-MS method [53].

The phenolic compounds with the most abandoned secoiridoids with high antioxidant potential present in olive mill effluents from Slovenian Istria can be a good source of high value compounds with beneficial effects on health as preservative and supplements in the food industry, pharmaceuticals and cosmetics (production of sunscreen and treatment of skin related disorders) [54,55].

## 3. Materials and Methods

### 3.1. General Experimental Procedures

The pomace samples from different cultivars such as “Maurino”, “Leccino”, “Buga” and “Istrska belica” were freeze dried (Büchi 1-4 LC plus, Martin Christ, Osterode am Harz, Germany), following solvent removal in vacuo (Büchi Rotavapor R-300 Dynamic, Martin Christ, Osterode am Harz, Germany). The phenolic compounds were analyzed and characterized using an ultrahigh-pressure liquid chromatography system (HPLC; Agilent 1290 Infinity2 HPLC modules, Agilent, Santa Clara, CA, United States), interfaced with a qTOF mass spectrometer (ESI-QTOF; 6530 Agilent Technologies, Agilent, Santa Clara, CA, United States). The HPLC equipment incorporated a Poroshell 120 column (EC-C18; 2.7 µm; 3.0 × 150 mm; Agilent, Santa Clara, CA, United States). The radical scavenging activity was measured using the DPPH assay and determined at 515 nm using a microplate reader Infinite F200 (Tecan, Männedorf, Switzerland).

Analytical standards such as oleuropein (12247-10MG, Sigma Aldrich, Dermstadt, Germany), hydroxytyrosol (SI-H4291-25MG, Sigma Aldrich, Dermstadt, Germany), tyrosol (AL-188255-5G, Sigma Aldrich, Dermstadt, Germany), luteolin (SI-L9283-10MG, Sigma Aldrich, Dermstadt, Germany), verbascoside (V4015-10MG, Sigma Aldrich, Dermstadt, Germany) and apigenin (SI-SMB00702-5MG, Sigma Aldrich, Dermstadt, Germany) were used for quantification of phenolic compounds; siringic acid (S6881-5G, Sigma Aldrich, Dermstadt, Germany) was used as an internal standard; 2,2-Diphenyl-1-picrylhydrazyl (D9132-250MG, Sigma Aldrich, Dermstadt, Germany) was used for determination of radical scavenging activity for pomace extracts.

### 3.2. Samples

A total of 18 pomace (that amount to 90 L) samples of olives from *O. europaea* L. were collected weekly from the beginning of olive oil production until the end of the mill production season in 2018 and 2019 (14 October 2018–18 November 2018 and 16 October 2019–09 November 2019). During the 2018 growing season, the samples were collected from two olive mills, Franka Marzi and Lisjak (Koper, Slovenian Istria, Slovenia), using different processing technologies (two-phase—Pieralisi FP60 RS ATEX and three-phase decanter centrifuge—Alfa Laval × 4); in 2019, the samples were collected only from the three-phase decanter centrifuge (Franka Marzi). During the two-phase decanting process, olives are initially washed, crushed and malaxed (churned), and water is added to a horizontal centrifuge (40–60 L/100 kg fruits weight), separating pomace from the oily must consisting of the vegetable water and oil. Olive oil, pomace and wastewater are the final products formed at one end of the three-phase decanter. In contrast to three-phase decanter, the two-phase decanter requires no additional water due to the much higher centrifugal speeds, resulting in olive oil and wet olive cake or pomace [11,12,13].

This sampling strategy facilitated investigation of the variation in phenolic compound composition across a number of different olive *O. europaea* L. cultivars (“Maurino”, “Leccino”, “Buga” and “Istrska belica”), which reached maturity at different times during the growing season. The samples were composed of only one variety or mixed varieties. In addition to pomace samples, OMWW was also sampled from the mill using three-phase centrifugation. In contrast to the pomace samples, quantification of the phenolic compounds in OMWW samples from the three-phase decanter was not performed due to the unknown exact addition of tap water that varied from 10–25%.

The pomace samples were immediately freeze dried (Alpha 1–4, Martin Christ Buchi, Osterode am Harz, Germany) after sample collection directly from the running olive mill. The OMWW was frozen immediately after sample collection. Dry pomace and OMWW samples were stored in a freezer (−18 °C) on average three months prior to analysis.

#### Extraction of Phenolic Compounds

Phenolic compounds were extracted from freeze dried pomace (2 g) in methanol/water 80:20 (50 mL, pH 2-HCl) for 30 min with stirring at room temperature and then re-extracted with fresh solvent (20 mL) for 15 min. The combined extracts were filtered and defatted using hexane (30 mL × 2). The defatted extracts were filtered and concentrated in vacuo (1.5 h). The residue was reconstituted to 10 mL of methanol and re-filtered through a 0.2-µm plastic non-sterile filter. The procedure is described in detail elsewhere by Obied et al. [50].

The phenolic compounds from olive mill water (15 mL, Batch 4, Franka’s olive mill) were defatted using hexane (15 mL). The sample was further extracted with ethyl acetate (15 mL × 3) and then centrifuged (40,000 g, 15 min) and concentrated in vacuo. The residue was reconstituted with methanol (10 mL) and then diluted 10 times. The samples were filtered through 0.2 µm 0.2 PA (nylon) filters. The procedure is described by Obied et al. [50].

### 3.3. Determination of Phenolic Compounds by HPLC-DAD-ESI-TOF

Phenolic compounds were characterized by HPLC-ESI-QTOF-MS. An elution gradient of 100% water/formic acid (99.5: 0.5, *v*/*v*) (A) towards 100% acetonitrile/methanol (50:50, *v*/*v*) was used over a period of 20 min (flow rate: 0.5 mL min; injection volume: 1 μL, column temperature 50 °C) starting at 3.0% B and increased to 100.0% B in 15 min and held for 5 min. A more detailed procedure can be found in the IOC method (COI/T.20/Doc. No. 29, 2009) [56]. The separated phenolic compounds were first monitored using a diode-array detector (DAD) (280 nm), and then MS scans were performed in the *m*/*z* range 40–1000 (capillary voltage, 2.5 kV; gas temperature 250 °C; drying gas 8 L/min; sheath gas temperature 375 °C; sheath gas flow 11 L/min). The analyses were performed in negative ionization mode. In those conditions, the instruments were expected to provide experimental data with accuracy within ± 3 ppm. Automated MS/MS data-dependent acquisition was performed for ions detected in the full scan above 2000 counts with a cycle time of 0.5 s, a quadrupole isolation width in narrow ~1.3 Da, using the following collision energies of 10, 20 and 40 eV and a maximum of three selected precursor ions per cycle. The instrument was tuned in the low mass range (up to 1700 *m*/*z*) and in extended dynamic range (2 GHz) in negative mode. All data were processed using Qualitative Workflow (version B.08.00) and Qualitative Navigator software (version B.08.00).

The extracts were screened for the range of phenolic compounds previously reported in *O. europaea* L. [19,23,34,35,38], and their identification confirmed, based on accurate mass and fragmentation profile with literature data and analytical grade standards (hydroxytyrosol, luteolin, verbascoside, apigenin, oleuropein). While tyrosol cannot be detected by MS because of its high ionization energy, its presence in the extracts was confirmed by comparison with the retention times of the tyrosol standard solution using a DAD.

The quantification was performed using calibration graphs prepared using six commercial standards (oleuropein, hydroxytarosol, tyrosol, luteolin, verbascoside, apigenin) by HPLC-DAD and HPLC-ESI-QTOF. Oleuropein and other secoiridoids were quantified with the calibration curve of oleuropein; hydroxytyrosol and hydroxytyrosolhexose isomers with the calibration curve of hydroxytyrosol; tyrosol and tyrosol glucoside were quantified with the calibration curve of tyrosol; apigenin and apigenin derivates were quantified with the calibration curve of apigenin; luteolin and other flavonoids were quantified with calibration curve of luteolin and verbascoside with the calibration curve of verbascoside [23]. The calibration plots indicated good correlations between peak areas and commercial standard concentrations. Regression coefficients were higher than 0.990 (5 point per calibration graphs). The solution of syringic acid was added to the samples as an internal standard before phenolic compound extraction. The response factor of syringic acid was calculated according to the IOC method (COI/T.20/Doc. No. 29, 2009) [56] with no greater than ±8% deviation. LOQ was determined as the signal-to-noise ratio of 10:1 and varied in the range from 2 mg/kg to 12 mg/kg dried pomace sample. The standard deviation between duplicates was less than 5%.

### 3.4. Radical Scavenging Activity Measured using DPPH Assay

The antioxidant activity of the different extracts was assessed using the radical-scavenging ability in the 1,1-diphenyl-2-picrylhydrazyl (DPPH) radical assay and conducted as reported by Žegura et al. [57] with modifications, including replacement of methanol with ethanol and use of tyrosol, rather than ascorbic acid, as a standard for positive control.

Reaction mixtures containing 100 µL of differently diluted extracts and 100 µL 0.2 mM DPPH in methanol were incubated for 60 min in darkness at ambient temperature, using 96-well microtiter plates. The decrease in absorbance of the free radical DPPH was measured at 515 nm with a microplate reader. The free radical scavenging activity was calculated as the percentage of DPPH radical that was scavenged and details are explained elsewhere [51]. EC50 values concentration at 50% of DPPH radical scavenged were determined graphically from the curves. Two independent experiments with two replicates each were performed.

### 3.5. Statistical Analysis

All the data obtained were analyzed using STATA13/SE software (version 13). The normality of variable distributions was assessed using the Shapiro–Wilk test. Spearman Rank correlation was used for bivariate comparison of the content of phenolic compounds and EC50 (Table 1). The Wilcoxon–Mann–Whitney test was applied for comparison of two different groups. The level of statistical significance was set to *p* < 0.05.

## 4. Conclusions

This study reports, for the first time, that the technological approach used in olive mills to separate different fractions as a critical factor in determining the types and levels of phenolic compounds obtained in the resultant pomace in Slovenia. From this study, it is clear that the observed variations in the content of phenolic compounds between each sample are highly dependent on the olive mill separation technology used. Along with the potential to reduce the environmental burden of olive processing, by minimizing the amount of water required, this information is important from a techno-economic planning perspective and will inform the future development of olive biorefineries in Slovenia that link to a value chain of bio-based products, including phenolic compounds. The data presented provide a good platform for understanding the influence of the separation technologies used during olive oil production on the type and quantity of phenolic compounds found in the resultant OMWW and pomace. The upstream optimization and/or reconfiguration of current olive crop processing systems could result in isolation of enhanced levels of key phenolic molecules during downstream processing, in order to improve valorization opportunities for these underutilized by-product streams.

## Figures and Tables

**Figure 1 molecules-26-00007-f001:**
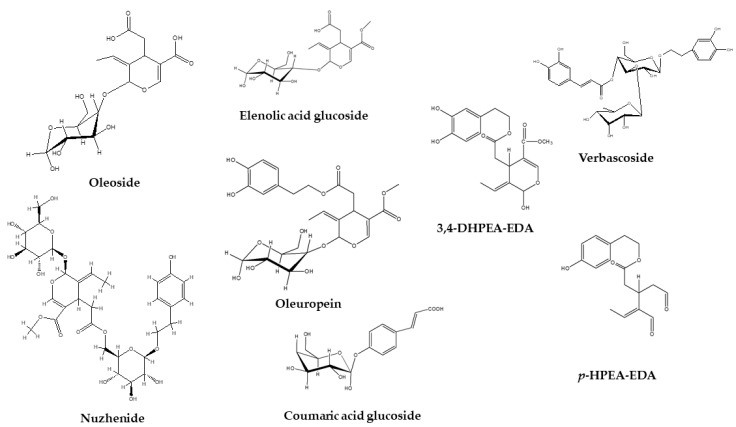
Phenolic compounds identified only in olive pomace and not in olive mill wastewater.

**Figure 2 molecules-26-00007-f002:**
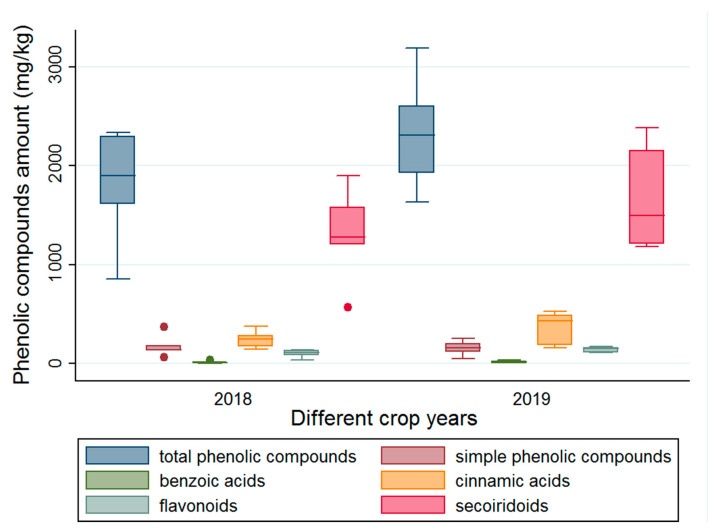
Total phenolic compounds, benzoic acids, flavonoids, simple phenolic compounds, cinnamic acids and secoiridoids determined according to crop years 2018 and 2019.

**Figure 3 molecules-26-00007-f003:**
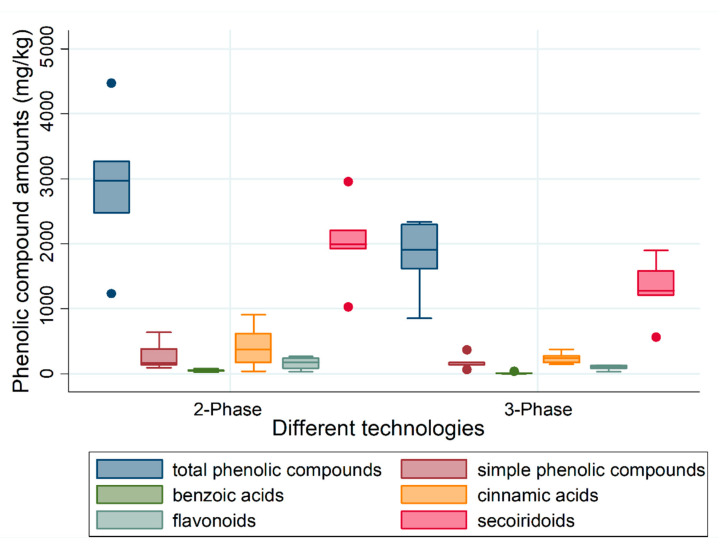
Total phenolic compound and phenolic compound composition according to technology used (two-phase separating decanter and three-phase separating decanter).

**Table 1 molecules-26-00007-t001:** Mean ± standard deviation (STD), median, minimum and maximum levels, total phenolic compounds, simple phenolic compounds, benzoic acids, cinnamic acids, flavonoids, secoiridoids in all pomace samples and Spearman Rank correlation coefficients between the determined levels of phenolic compounds and radical scavenging activity measured using 1,1-diphenyl-2-picrylhydrazyl (DPPH) assay.

Name of Compound	Mean ± STD (*n* = 18)	Median (*n* = 18)	Min (*n* = 18)	Max (*n* = 18)	r_s_ DPPH corr. sig. *p* < 0.05
Oleoside 1 (mg/kg d.w.)	33 ± 22	26	13	90	−0.77
Oleoside 2 (mg/kg d.w.)	27 ± 14	30	<LOQ	46	
Hydroxytyrosol, hydroxytyrosol glucoside, Oleoside 3 (mg/kg d.w.)	157 ± 133	115	45	605	−0.70
Elenolic acid glucoside 1 (mg/kg d.w.)	16 ± 21	11	<LOQ	76	−0.67
Elenolic acid glucoside 2 (mg/kg d.w.)	3.1 ± 6.6	<LOQ	<LOQ	24	
Elenolic acid glucoside 3 (mg/kg d.w.)	50 ± 35	48	<LOQ	136	−0.66
Tyrosol (mg/kg d.w.)	37 ± 30	30	<LOQ	133	
Sacolagonoside (mg/kg d.w.)	106 ± 60	98	19	274	
Trans *p*-coumaric acid 4-glucoside (mg/kg d.w.)	53 ± 56	41	<LOQ	150	
Caffeic acid (mg/kg d.w.)	26 ± 30	12	<LOQ	97	−0.63
Elenolic acid glucoside 4 (mg/kg d.w.)	39 ± 45	14	<LOQ	126	
Luteolin-4′,7-*O*-diglucoside (mg/kg d.w.)	4.8 ± 16	<LOQ	<LOQ	67	
β-OH-verbascoside 1 (mg/kg d.w.)	9.1 ± 13	<LOQ	<LOQ	44	
β-OH-verbascoside 2 (mg/kg d.w.)	64 ± 41	64	<LOQ	137	−0.67
Vanillin (mg/kg d.w.)	23 ± 22	16	<LOQ	74	−0.67
Verbascoside 1 (mg/kg d.w.)	76 ± 81	60	<LOQ	261	
Dimethyloleuropein (mg/kg d.w.)	31 ± 77	<LOQ	<LOQ	284	
Rutin (mg/kg d.w.)	48 ± 42	39	16	204	
Verbasciside 2 (mg/kg d.w.)	112 ± 112	84	<LOQ	405	
Luteolin-7′-*O*-glucoside (mg/kg d.w.)	9.5 ± 18	0	<LOQ	47	
Luteolin rutinoside (mg/kg d.w.)	34 ± 41	20	<LOQ	123	
Nuzhaenide 1 (mg/kg d.w.)	34 ± 45	14	<LOQ	146	
Luteolin-4′-*O*-glucoside (mg/kg d.w.)	14 ± 19	0.1	<LOQ	58	
Caffeoyl-6-secologanoside (mg/kg d.w.)	31 ± 77	<LOQ	<LOQ	285	
Nuzhenide 2 (mg/kg d.w.)	194 ± 176	123	<LOQ	551	
Luteolin-3′-*O*-glucoside (mg/kg d.w.)	17 ± 21	7.8	<LOQ	69	
3,4-DHPEA EDA/Oleuroside 2 (mg/kg d.w.)	985 ± 510	985	293	1981	−0.60
Oleuropein aglycone 2 (mg/kg d.w.)	14 ± 59	<LOQ	<LOQ	248	
Oleuropein/Oleuroside 3 (mg/kg d.w.)	8.9 ± 17	<LOQ	<LOQ	55	
Ligstroside (mg/kg d.w.)	18 ± 42	<LOQ	<LOQ	162	
Oleuropein aglycone 3 (mg/kg d.w.)	19 ± 35	<LOQ	<LOQ	128	
*p*-HPEA-EDA (mg/kg d.w.)	13 ± 25	<LOQ	<LOQ	91	
Oleuropein aglycone 5 (mg/kg d.w.)	0.9 ± 3.7	<LOQ	<LOQ	16	
Apigenin (mg/kg d.w.)	7.0 ± 5.2	5.8	<LOQ	20	−0.66
Oleuropein aglycone 7 (mg/kg d.w.)	18 ± 38	0	<LOQ	154	
3,4-DHPEA EDA (mg/kg d.w.)	7.4 ± 17	0	<LOQ	52	
Oleuropein aglycone 8 (mg/kg d.w.)	9.7 ± 9.0	12	<LOQ	30	
Oleuropein aglycone 9 (mg/kg d.w.)	0.7 ± 3.0	<LOQ	<LOQ	13	
Radical scavenging activity by DPPH EC50 (µg/mL)	414 ± 242	317	200	1060	
Simple phenolic compounds (m/kg d.w.)	194 ± 141	154	45	637	−0.71
Benzoic acids (mg/kg d.w.)	23 ± 22	16	<LOQ	74	−0.67
Cinnamic acids (mg/kg d.w.)	340 ± 220	265	36	905	−0.60
Flavonoids (mg/kg d.w.)	134 ± 61	129	31	266	
Secoiridoids (mg/kg d.w.)	1657 ± 582	1632	564	2953	−0.72
Total phenolic compounds (mg/kg d.w.)	2348 ± 849	2317	851	4473	−0.81

## Data Availability

Samples of the compounds are not available from the authors.

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
