# Peer review of "Developing an Olive Biorefinery in Slovenia: Analysis of Phenolic Compounds Found in Olive Mill Pomace and Wastewater"

_molecules, 2020, doi:10.3390/molecules26010007_

Round 1
Reviewer 1 Report
The authors have put a lot of work to improvment the manuscript, and at present it is much better. Its layout is correct and the information presented has a greater utilitarian value. The scientific value of the manuscript may be questioned due to the lack of reproducibility of the experiments and the enormous number of variables that influenced the results obtained. Conclusions drawn on the basis of the obtained results are very general, not repeatable and not always reliable.
On the other hand, in some ways the advantage of the manuscript is that the research was done on a large and actual scale. This is a tip for operators and help for the implementation processes. Due to this fact, I ultimately believe that the manuscript may be published.
My comments:
The authors conducted the research during two seasons 2018-2019. In studies related to agronomy, reliable results are obtained only after at least three growing seasons. Can the authors comment on this?
Figure 3 needs to be corrected. Something went wrong with the Y axis.
Reviewer 2 Report
Thank you good extensive editing and big improvement than the previous one.
Good luck
Author Response
Dear Reviewer,
we found Your suggestions and comments very useful and helpful for improvement of the quality of our manuscript.
With kind regards,
Your sincerely,
Dr. Matthew Schwarzkopf
Reviewer 3 Report
The revised manuscript entitled as “ Developing an olive biorefinery in Slovenia: Analysis of phenolic compounds found in olive mill pomace and waste water” is certainly an improved version of the work previously submitted by the same authors.
However, there are still several minor but also critical points that should be carefully re-considered and revised in order the manuscript to become suitable for publication:
L. 134-137. the discussion points are contradictory, you write that HT is not detected by MS and then that it was confirmed by mass spectraL. 149 what do you mean "only in their structure?"
L.160. delete "while"
Table 1 legend.
-write "mean ± standard deviation"
-delete "each determined phenolic compound".
-refer to what exactly is shown in the last column, abbreviations, meaning etc, or else, transfer to the text. The information is still confusing
-refer to the raw material tested
L. 302-304. see comments on the Table 1 legend.
L. 314. check throughout the format of given values in range e.g. 71 ± 7%
L. 318-323. how about post-harvest treatments?
L. 332 secoiridoids
L. 342. delete "of variation"
L. 346. re-phrase "the two factors discussed.."
L. 421-424. This part needs further discussion and clear reference to Table 1 data.
L. 424-427. so why did you make these calculations in your study?
L. 435. Please refer to the type of cultivar tested, here.
L. 465-466. What was the the botanical source of the test samples? were they mixtures of all? the botanical source is highlighted in the Abstract and the Introduction sections but not discussed or refered to here
L. 468-469. do you mean OMWW from three-phase mills???
L. 473-474. indicate the storage time before analysis
L. 489. check the format 99,5:0,5
L. 492. Please report the code of the standard method
L. 507-508. Check the relevant point in your introduction L. 134
L. 510-518. How many points per calibration graph? Please indicate this
L. 520-521. Here you refer to the reproducibility of the extraction procedure, not only the HPLC instrumental one. LOQ of syringic acid, only? What was the recovery rate? You should check this with representatives of each class of compounds you tested
L. 524 the word "assessed" is more accurate than measured
L. 526. I don't really agree that changing the solvent or the reference compound is minor modification, justify in this case.
L. 559. Maybe this is the first time for studying the waste products from Slovenian olive mills..
L. 570. the study of cultivar type effect is not reflected in the presented results of this manuscript
Round 2
Reviewer 3 Report
I belive the manuscript has been improved and now warrants publication in Molecules
This manuscript is a resubmission of an earlier submission. The following is a list of the peer review reports and author responses from that submission.
Round 1
Reviewer 1 Report
Manuscript ID molecules-954324 presents research on phenolic compounds content in olive mill effluents. The topic undertaken by the authors is interesting and important. However, in my opinion, the presented results do not bring anything new to the knowledge that is already commonly known. In many already published scientific papers, it can be find a detailed description of types of phenolic compounds, variation in the compounds and amount of phenolic compounds that are extracted from olive mill effluents. For this reason, I believe that the presented results are only a repetition of similar, already completed research and therefore are not original and innovative.
In my opinion, the manuscript is written in a not very scientific way. The Authors merely mention and briefly describe the identified phenolic compounds. Taking into account the rich literature in this field, the work lacks an in-depth, reliable discussion and comparative analysis in the results obtained by other Authors. And the manuscript also failed to analyze the impact of the identified phenolic compounds on the possibilities and limitations in the management and use of olive mill effluent. The scientific discussion should be extended and the results obtained should be assessed against those of other Authors.
There are numerous minor English language errors which detract from the content. These must be corrected. While it is important to compare and benchmark presented results to that of others Authors results are currently getting lost in the extensive discussion in the results and discussion section. Please make clear what is novel and what it has added. Conclusions need to be more than a statement of results. You need to broaden your conclusions to state the implications for others in their work and remember the scientific process is focused around: hypothesis, experiment, results, conclusion.
In my opinion, the manuscript ID molecules-954324 should not be published in its current form. It requires deep changes and additions and should be re-submitted in a new version.
Reviewer 2 Report
- Very good detail analysis of phenolic compounds in OMWW and pomace
- High Variation with results (Min - Max) is this for 18 samples what is the the coefficient of variation for Each for each compound.
- More updated new references should be included
- Three phase pressing (OMWW) contains more phenol because the phenol soluble in water and has more contact time in three phase mill
- What about Phenol concentration in Pomace from 3-phase mill and Alperjuo (The study should investigate this for comparison because Alberjo water content from the 2-phase mill is much higher than the pomace form 3-phase mill). Each has its own advantages/disadvantages
- so what are the commercial values of these compounds, how much the cost of extraction per liter and the expected market : Pharmatical, food industries, dieting and nutrition values and so on.
- Seasonal Variation is not significant nor necessary. Huge quantities of waste is produced after each milling season
Reviewer 3 Report
The manuscript entitled as “Developing an olive biorefinery in Slovenia: Analysis of phenolic compounds found in olive mill effluents” presents data about variations in phenolic composition of selected Slovenian olive mill effluents according to processing (decanting system) or seasonal changes (two consecutive years). The authors recognise that their results are preliminary and a larger sample size is needed to understand the annual variations (L. 262-263) which could be emphasized also in the abstract. In general, the concept is good but the analytical strategy is not well defined. The relevant literature (e.g. refs no 15-19) has not been studied in-depth and discussed appropriately throughout the text. The authors should justify why they chose the exact analytical conditions for the recovery of phenols from the specific matrices and also, provide minimum performance characteristics of the method used for the subsequent chromatographic separation and detection. It is not clear whether HPLC-DAD and HPLC-MS analyses were on-line or not, also what were the instrument type, column characteristics etc. Their Introduction part should highlight possible analytical difficulties encountered in literature regarding the identification of individual phenols, complexes and derivatives in olive products. Also, the possible storage effect is not addressed at all in this manuscript. To my mind, the discussion about DPPH data is not insightful and does not add any new and interesting point so it could be excluded.
The overall quality is poor from an analytical point of view, more details are needed to ensure the validity of measurements. Therefore, I think that the manuscript does not deserve publication in “Molecules” unless major revisions are made.
Some more specific comments are given below
L. 43-45 in comparison to what?
L. 80-81. within a short period of time?
L. 99-100. The legend is not self-explanatory regarding DPPH results.
L. 94-203-. please refer to peak numbers, as shown in the supplementary chromatogram or Table, not just RT values e.g. Peak no 17 was eluted at 8.2 min and assigned to a verbasoside isomer according to UV and MS-fragmentation pattern. Also, discuss in terms of compound abundancy. What were the major constituents in these extracts?
L. 139. Delete “in structure”. Do you mean that it was tentatively identified as…?
L. 144. used by whom?where?
L. 145-148. This part has to be expanded to contain essential performance characteristics other than the overall time of analysis. Chromatographic resolution, precision/recovery, LOD/LOQ, possible artefacts etc should be discussed for targeted profiling purposes
L. 152-153. this is not the proper way to report the chromatographic results. Please indicate at least the relative retention time.
L. 157. observed by whom/when/where?
L. 158. secologanoside structure is missing
L. 212-214. misleading sentence, please re-write so as to clarify what is reported in literature and what is your study's outcome.
L. 220-222. why is that? Do you think it might be reproducible under different analytical conditions?
L. 229-231. the sentence seems irrelevant with the abovementioned.
L. 269. please re-write the legend to specify the samples tested
L. 285-287. Why do you think only vanillin recovery is favoured?
L. 296-307. I do not understand what is the point of using this test, after all. It is well accepted that the DPPH results are prone to various interpretations according also to the analytical assay conditions. It seems that it does not add any insight to the data reported here.
L. 328-330 What was the sampling procedure. How many samples per mill? is the volume processed by each mill representative of the local production?
L. 361. why did you choose this particular non-validated for olive-products LC protocol? what to you mean "to make the procedure applicable for different column dimensions"? the HPLC part is essential to be presented clearly. Specific method performance criteria have to be included in the study otherwise inter-laboratory data comparison is not feasible.
L. 383. added before phenol extraction?
L. 414. specify from which kind of mill
L. 456. This section is too big, please focus on the most relevant to the topic references